TECHNICAL RELEASE

# Trumpet plots: visualizing the relationship between allele frequency and effect size in genetic association studies

Lucia Corte[1,2], Lathan Liou[1], Paul F. O'Reilly[1] and Judit García-González[1,*]

1 Department of Genetics and Genomic Sciences, Icahn School of Medicine at Mount Sinai, New York City, NY 10029, USA

2 Center for Excellence in Youth Education, Icahn School of Medicine at Mount Sinai, New York City, NY 10029, USA

## ABSTRACT

Recent advances in genome-wide association and sequencing studies have shown that the genetic architecture of complex traits and diseases involves a combination of rare and common genetic variants distributed throughout the genome. One way to better understand this architecture is to visualize genetic associations across a wide range of allele frequencies. However, there is currently no standardized or consistent graphical representation for effectively illustrating these results.

Here we propose a standardized approach for visualizing the effect size of risk variants across the allele frequency spectrum. The proposed plots have a distinctive trumpet shape: with the majority of variants having high frequency and small effects, and a small number of variants having lower frequency and larger effects. To demonstrate the utility of trumpet plots in illustrating the relationship between the number of variants, their frequency, and the magnitude of their effects in shaping the genetic architecture of complex traits and diseases, we generated trumpet plots for more than one hundred traits in the UK Biobank. To facilitate their broader use, we developed an R package, 'TrumpetPlots' (available at the Comprehensive R Archive Network) and R Shiny application, 'Shiny Trumpets' (available at https://juditgg.shinyapps.io/shinytrumpets/) that allows users to explore these results and submit their own data.

**Submitted:** 20 April 2023

\* Corresponding author. E-mail: judit.garciagonzalez@mssm.edu

Preprint submitted at https://doi.org/10.1101/2023.04.21.23288923

**Subjects** Genetics and Genomics, Bioinformatics, Statistics and Probability

## STATEMENT OF NEED

Visualizations are powerful tools that have helped the field of genetics to better understand and communicate complex findings. By using visual aids like Manhattan plots and volcano plots, researchers can more easily pinpoint genetic variants identified through genome-wide association studies (GWAS). With the advancement of GWAS and sequencing studies, a mounting number of significant genetic variants – both common and rare – are being discovered. To better understand the relationship between these variants, combining these findings into single visualizations helps to observe the relationship between effect size and allele frequency, providing a clearer picture of the genetic architecture of different traits and diseases. However, *there is currently no consistent method for illustrating such results.* In this paper, we propose a standardized approach for visualizing the effect size of risk variants across the allele frequency spectrum, generate plots for over a hundred traits

in the UK Biobank, and provide to the field an R package and R Shiny application to allow users to explore their own results.

## BACKGROUND

Results visualization is an essential tool for interpreting complex data. By using visual representations such as graphs, charts, and plots, researchers can quickly identify patterns, trends, and outliers that may not be detected in tables of raw data. Visualizations help researchers gain a more intuitive and comprehensive understanding of the data, especially when dealing with large and complex data sets. Furthermore, visualizing results can facilitate communication of research findings to a broader audience, including non-experts [1]. Visual representations are often more accessible and engaging, making it easier for others to understand and appreciate the significance of the research conducted.

In the field of genetics, the use of visualizations has revolutionized the interpretation and communication of research findings. Over the past two decades, visualizations such as Manhattan plots [2] which display the results of GWAS, software packages like haploview [3] that analyze and visualize linkage disequilibrium (LD) patterns of GWAS-associated loci, and volcano plots that assess patterns of differential gene expression [4] have all played crucial roles in illustrating and sharing the key summaries of data that have advanced the field. These and other visualizations [5] have allowed the genetics field to more easily identify candidate causal variants, relevant genes, and potential outliers that may not be apparent in tables of GWAS summary statistics or differential expression results.

More recent advances in GWAS and sequencing studies [6–8] have resulted in the identification of an increasing number of significant genetic variants, including both common [8] and rare variants [5, 6]. Researchers are now starting to combine these findings into single visualizations to observe the relationship between effect size and allele frequency across the full range of significantly associated variants. Given that the number of risk-conferring variants, their frequency in a population, and their effect size can vary across different traits and diseases, using these plots can provide a better and instantaneous understanding of their relative genetic architecture. Recent studies on height [9], schizophrenia [10] and coronary artery disease [11, 12] have already included this full range as the main figure, highlighting the utility of this type of visualization. However, a formal and consistent method for illustrating these results has not yet emerged.

The aim of this work is to introduce an R package and R Shiny application to illustrate the distribution of risk variants across a wide range of allele frequencies. We term the resulting plots 'trumpet plots', due to their trumpet-like shape. To demonstrate their utility, we generated trumpet plots for over one hundred continuous traits available in the UK Biobank [13], illustrating the distribution of risk variants across an effect allele frequency range between 0.00001 and 1. These plots are available at https://juditgg.shinyapps.io/shinytrumpets/, and we illustrate a single trumpet plot combining the results of all of these in Figure 1.

We propose that trumpet plots are valuable representations of genetic associations across the full allele frequency spectrum that can help researchers to better understand the genetic architecture of traits and diseases, and potentially aid in study design and the prioritisation of investments to discover new variants that contribute to disease.

## METHODS

In the following sections, we will explain the various decisions we made when creating trumpet plots. These decisions include selecting the appropriate scale to represent allele frequencies, deciding whether to use the full GWAS summary statistics or independent GWAS variants, addressing issues related to the reporting of rare-variant association tests, determining whether to include power curves, and considering the effect size sign of the variants included. By carefully considering each of these factors, we aimed to create informative and visually appealing trumpet plots that illustrate the effect size of genetic associations across a wide range of allele frequencies.

### Using the logarithmic vs linear scale to represent allele frequencies

In the representation of allele frequencies, the range of values can vary greatly between the smallest and largest frequency. When these associations are plotted on a linear scale, rare variants can be obscured or difficult to distinguish. To address this issue, we recommend using a logarithmic (log) scale – we use log base 10 – for allele frequencies in trumpet plots. Compared to a linear scale, the log scale uses increments that represent a relative increase or decrease, rather than a fixed-value increase or decrease. The log scale compresses the allele frequencies that are most common, which results in a more even distribution of values across the scale. This scale of visualization facilitates the identification of important patterns and trends.

### Identification of independent significant variants to enhance the interpretation of trumpet plots

Genetic association studies involve testing up to millions of genetic variants for their association with a particular trait or disease. However, many of these variants are correlated with each other due to their physical proximity on the genome, which is known as LD. This means that many variants nearby to causal variants often show significant associations with the trait under study, due only to their correlation rather than to any biological involvement with the trait.

Two methods that can be used to identify independent significant variants in a GWAS are clumping and conditional analysis [14, 15]. Clumping involves selecting a subset of independent significant variants by choosing a lead variant for each LD cluster and then discarding all other variants in that cluster. The lead variant is typically the one with the strongest association with the trait of interest. Conditional analysis, on the other hand, involves identifying independent significant variants after performing a joint analysis of multiple variants together. In this approach, the effect of one variant is *conditioned* on the effect of other variants; that is, the association between the trait of interest and one variant is evaluated after accounting for the effect of other variants. This can either be performed as a joint analysis (e.g., a regression), with multiple variants in one model, or else as an iterative process, where the lead variant and each other variant in the region are tested jointly, one by one. By considering the effects of multiple variants jointly in single models, conditional analysis helps to identify independent signals more accurately than clumping – which relies on only correlations between variants to indirectly infer independent signals.

Since the correlation between variants can make it challenging to interpret trumpet plots, we recommend plotting only independent significant variants: variants that represent distinct genetic associations with the trait of interest.

### Variant-level associations for low allele frequencies

While GWAS are a valuable tool for detecting common genetic variants associated with complex traits or diseases, they have limited power to identify associations with rare variants. To address this limitation, sequencing studies [16–18], such as whole-exome sequencing and whole-genome sequencing, have been used to detect variant-level associations with low allele frequencies.

To further enhance the statistical power of rare-variant associations, a commonly used strategy is to aggregate the rare variants detected into functional genetic units, such as genes, and perform collective variation analysis (e.g., gene burden tests [19–21]). However, we find that the reporting of rare-variant association analyses varies across studies. Some studies report results at the variant level, while others report only results of the functional unit in which rare variants were aggregated. This makes comparing results across independent studies, and determining the functional significance of specific variants, challenging.

We therefore encourage the reporting of results at the variant level, so that they can be included in visualizations of allele frequency in relation to effect size, such as trumpet plots, aiding study comparisons and variant interpretation. Although the analysis of genetic variants at low frequency is expected to improve with the availability of biobank-scale samples and the development of new methods to reduce biases in association tests, caution should still be exercised when inspecting associations with rare variants, as these can suffer from instability and low power, particularly in relation to binary traits.

### Statistical power considerations

GWAS requires careful consideration of statistical power [22], which depends on various factors, including allele frequency and the effect size of variants – represented by the *x*- and *y*-axes of trumpet plots, respectively. Common variants – usually defined as having allele frequency greater than 1% – tend to have higher power in association studies because common causal variants are more likely to be present in the sample (either genotyped or imputed), and because their relatively balanced number of alleles is akin to having a larger sample size. Variants with larger effect sizes have higher power because their effects are further from the null hypothesis of zero effect.

We therefore recommend incorporating power curves into trumpet plots, since they visually represent the statistical power across the spectrum of allele frequency for a given sample size and effect size [23]. Moreover, power curves can aid in identifying parts of the association testing space in which the power to detect significant associations is low.

### Two alternative approaches to illustrate the joint distribution of allele frequency and effect size

One approach to illustrate the relationship between allele frequency and effect size is to plot only positive effects: that is, the allele effect for each variant that increases the value of the phenotype. In this case, the effect sizes are always positive, and both the allele and sign of the association regression coefficients (betas or odds ratios) need to be flipped (to the other allele) if they are reported as negative, to ensure that the effect size is greater than zero. If this 'flipping' is required, then the allele frequency of the other allele should be reported, which will be 1 minus the original allele frequency. In this case, the allele frequencies of the plot range from 0 to 1.



The other approach, which we recommend, allows for both positive (risk allele in the context of disease phenotypes) and negative (protective allele in the context of disease phenotypes) effect sizes and always corresponds to the minor allele. In this case, the effect size of the allele can have either a positive or a negative value, and the allele frequencies of the plot range from 0 to 0.5.

## Practical example: Generating trumpet plots for 129 traits in the UK Biobank

We examined all continuous UK Biobank traits with available GWAS analyses performed by Benjamin Neale's group [24], and searched to confirm whether rare-variant associations were available for the same trait (by UK Biobank Field ID) in the exome sequencing analysis performed by the Regeneron team [13].

Common variant associations were extracted from the Neale group's GWAS summary statistics. For each GWAS, we extracted the independent variants using COJO GCTA (–cojo-slct command), and a random subset of 4,000 unrelated individuals with European ancestry from the UK Biobank as an LD reference panel. We selected independent variants with minor allele frequency of >0.01 and association $P$-value < $5 \times 10^{-8}$ within a 100-Kb window.

Rare-variant association results were extracted from the supplementary data table 2 (SD2) of the Regeneron study [13]. This study reported results for burden tests (which typically aggregated variants and indels) and individual rare-variant–level tests. To ensure that the effect sizes reported in our analyses corresponded with individual rare variants, we extracted results for only 'singleton variants' with predicted loss of function – including stop-gain, frameshift, stop-lost, start-lost and essential splice variants – and deleterious missense variants.

We utilized our R package, TrumpetPlots [25] (RRID:SCR_023742) to create plots depicting the relationship between allele frequency ($x$-axis) and odds ratio ($y$-axis). Figure 1 illustrates the combination of results from 129 continuous traits, aggregated into a single trumpet plot. When all the traits are collectively represented, a change in the number of associations around allele frequency of 0.01 is observed. This is likely due to a combination of factors, including differences in genome coverage, quality control and statistical power of the two studies used. For allele frequencies of >0.01, association results were extracted from GWAS that used genotyping arrays to assess genotyped variants across the entire genome. In contrast, for allele frequencies of <0.01, association results were extracted from exome sequencing of coding variants only, which constitute a small fraction of the genome. These variants were further filtered to include only rare, singleton variants with predicted loss of function; while these variants may be expected to have relatively large effect sizes, their statistical power to identify significant associations corresponding to variants of small effect, is substantially lower than that of common variants.

## R Shiny application

We developed a user-friendly web application called Shiny Trumpets to visualize trumpet plots for our UK Biobank results, as well as any other genetic association results that can be uploaded by the user. With Shiny Trumpets, researchers with no knowledge of R programming can easily upload and visualize their own data sets.

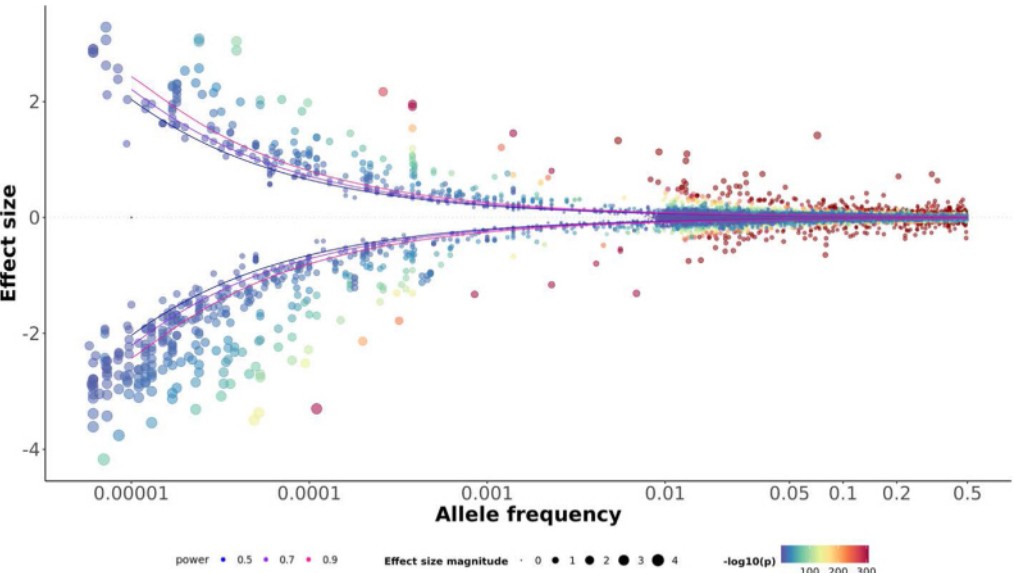

**Figure 1.** Distribution of allele frequencies and effect sizes for genetic associations across 129 continuous traits from the UK Biobank. Power curves for statistical power of 0.5 (blue), 0.7 (purple) and 0.9 (pink) were constructed for the median genome-wide association study sample size ($N$ = 351,550). The colours of the dots represent the $-\log_{10}$ of the association $P$-value. The size of the dot is proportional to the effect size (represented in the $y$-axis). figure-1.html

If a user uploads their own results, the Shiny Trumpets application prompts them to upload the input data files and specify the sample size used for the study, such as the GWAS sample size. This information is used to perform power calculations for the visualization. Shiny Trumpets offers an intuitive interface for users to explore and download trumpet plots.

## DISCUSSION

Visual representations of genetics and genomics results, such Manhattan plots [2], Q-Q plots [2], haploview [3] or volcano plots [4], have been helpful in interpreting research findings and in identifying patterns, trends, and outliers that may not be easily apparent in tables of raw data. These visualizations have revolutionized the interpretation and communication of research findings relating to the identification of GWAS-associated loci, putatively causal genes, and potential outliers.

In this manuscript, we introduce a new R package and R Shiny application to illustrate the distribution of risk variant effect sizes across a wide range of allele frequencies, which we coin: 'TrumpetPlots'. We illustrate the distribution of variant effect sizes across the allele frequency range (from 0.00001 to 1) for over 100 continuous traits available in the UK Biobank, and propose that these plots are valuable representations of genetic associations that can help researchers better understand the genetic architecture of traits and diseases and prioritize certain study designs (e.g., sequencing or GWAS) to discover new variants that contribute to disease.

Alternative combinations of the results from genetic association analyses can lead to various types of plots, each with distinct shapes that differ from a trumpet. For instance, Manhattan plots have gained popularity as a means of illustrating association results, and related variations like Miami plots [27] and Brisbane plots [9] have also been reported. In a

previous study [28], the idea of adapting volcano plots (commonly used to represent differential gene expression analyses) was proposed for genetic association studies. Other metrics – such as the proportion of variance explained, or the population attributable risk of each variant – could be represented in relation to the risk allele frequency [29]. All these metrics have different properties and assumptions, which can influence their use and interpretation [29]. For example, illustrating effect sizes is particularly suitable for identifying genetic variants with strong effects regardless of how common a variant is in the population. This is especially important for the discovery and prioritization of candidate genes, and to gain biological insights of the traits or disease under study. However, it also highlights the presence of large-effect rare variants that, due to their low frequency, may have a small contribution to population-level disease risk.

One important consideration when interpreting the trumpet plots we constructed for the UK Biobank is that they represent only individuals of European ancestry. The relationship between effect size and allele frequency can be affected by population genetic differences [30, 31] and, as such, one interesting application of trumpet plots could be to compare the joint distribution of allele frequencies and effect sizes across different ancestries to identify similarities and differences for further investigation. Insights about the similarities and differences across populations, in the relationship between effect size and allele frequency, could have important implications for disease risk prediction and prevention strategies.

In conclusion, we emphasize the significance of data visualization in the genetics field and present a novel R package and R Shiny application for visualizing the relationship between allele frequency and effect size in association studies. We hope that the proposed 'trumpet plots' will provide a valuable representation of genetic associations and enhance the interpretation of association results across the allele frequency spectrum.

## AVAILABILITY OF SOURCE CODE AND REQUIREMENTS

- Project name:

  - R package available in the Comprehensive R Archive Network https://cran.r-project.org/package=TrumpetPlots and in GitLab project 'TrumpetPlots' https://gitlab.com/JuditGG/trumpetplots
  - R Shiny app and analyses in the UK Biobank available in project '*freq_or_plots*' https://gitlab.com/JuditGG/freq_or_plots

- Project homepage: https://juditgg.shinyapps.io/shinytrumpets/
- Operating system(s): Platform-independent
- Programming language: R
- biotools ID: biotools:trumpetplots
- RRID:SCR_023742
- License: MIT.

## DATA AVAILABILITY

All data used in this manuscript is publicly available. Rare-variant associations are available in supplementary data table 2 of the original publication [13]; GWAS summary statistics are available on the website https://www.nealelab.is/uk-biobank/.

The code is freely available and open to others' contributions at https://gitlab.com/JuditGG/freq_or_plots (UK Biobank analyses),



https://gitlab.com/JuditGG/trumpetplots (R package with test data) and
https://juditgg.shinyapps.io/shinytrumpets/ (R Shiny application).

To seek support or to report issues, users can visit
https://gitlab.com/JuditGG/freq_or_plots/-/issues (for questions or issues related to the R
Shiny application) and https://gitlab.com/JuditGG/trumpetplots/-/issues (for questions or
issues related to the R package).

Snapshots of the code are also available from the GigaDB repository [26].

## LIST OF ABBREVIATIONS

GWAS: genome-wide association study; LD: linkage disequilibrium; log: logarithm or
logarithmic.

## DECLARATIONS

### Ethics approval and consent to participate

The authors declare that ethical approval was not required for this type of research.

### Competing Interests

The authors declare that they have no competing interests.

### Authors' contributions

LC: Data Curation, Formal Analysis, Investigation, Writing – Original Draft Preparation.
LL: Software, Validation, Visualization. PFO: Conceptualization, Funding Acquisition,
Formal Analysis, Supervision, Writing – Review & Editing. JGG: Conceptualization, Data
Curation, Formal Analysis, Investigation, Software, Validation, Supervision, Visualization,
Writing – Original Draft Preparation, Writing – Review & Editing.

### Funding

This work was supported by a grant from the National Institutes of Health (R01MH122866)
to PFO, by a 2022 NARSAD Young Investigator Grant (Number 30749) by the Brain &
Behavior Research Foundation to JGG, and through the computational resources and staff
expertise provided by Scientific Computing and the Data Ark (Data Commons) teams at the
Icahn School of Medicine at Mount Sinai.

### Acknowledgements

We thank the participants of the UK Biobank and the scientists involved in the construction
of this resource. We would like to thank Dr Shea Andrews for helpful discussions on several
aspects of the project. We would also like to express our gratitude to the Center for
Excellence in Youth Education (CEYE) program for their support and training, which
enabled us to carry out this research. Without the invaluable assistance and dedication of
CEYE staff, this project would not have been possible.

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
