## [Editor Report]

Editor’s AssessmentThis work presents a new standardized graphical approach for visualizing genetic associations across a wide range of allele frequencies. These proposed TrumpetPlots have a distinctive trumpet shape, hence the proposed name. With the majority of variants having low frequency and small effects, while a small number of variants have higher frequency and larger effects, this view can help to provide new and valuable insights into the genetic basis of traits and diseases, and also help prioritize efforts to discover new risk variants. The tool is provided as a novel R package and R Shiny application and to demonstrate its use the article illustrates the distribution of variant effect sizes across the allele frequency range for over 100 continuous traits available in the UK Biobank. After some problems in testing the package is now available and easy to deploy via CRAN.

---

## [Reviewer Report]

Comments on revised manuscriptI have read authors' response and I'm mostly satisfied. Only two minor comments:
* Witte 2014 Nature Rev. Genet. article summarizes the point I tried to make well. I understand that rare variants should have a relatively higher effect from an evolutionary perspective, but since these are rare, their individual or even collective contribution to a disease in the population is still small. A casual reader may not realize this point and I think it would be helpful to cite Witte's article.
* My minor comment on Fig.1 is still not addressed: there seem to be more points on the right side of p=0.01 line than the left side. Why this discontinuity? (the added text in Revision is about the color and size of the dots, not about this discontinuity)

---

## [Reviewer Report]

Reviewer name and names of any other individual's who aided in reviewerClara AlbiñanaDo you understand and agree to our policy of having open and named reviews, and having your review included with the published manuscript. (If no, please inform the editor that you cannot review this manuscript.)YesIs the language of sufficient quality?YesPlease add additional comments on language quality to clarify if neededIs there a clear statement of need explaining what problems the software is designed to solve and who the target audience is? YesAdditional CommentsIs the source code available, and has an appropriate Open Source Initiative license <a href="https://opensource.org/licenses" target="_blank">(https://opensource.org/licenses)</a> been assigned to the code?YesAdditional CommentsAs Open Source Software are there guidelines on how to contribute, report issues or seek support on the code?NoAdditional CommentsAlthough there are no explicit guidelines for contribution in the manuscript or website, it is true that by placing the project on gitlab it is possible to contribute to the project / open issues.Is the code executable?NoAdditional CommentsUnfortunately, I wasn't able to install the R package. I have now opened an issue on the gitlab page so that it can hopefully get solved.Is installation/deployment sufficiently outlined in the paper and documentation, and does it proceed as outlined?YesAdditional CommentsIt is very common for new R packages to just use devtools for installation.Is the documentation provided clear and user friendly?YesAdditional CommentsThe requirements for generating a trumpet plot just involve providing a set of GWAS summary statistics with column-specific names, together with the GWAS sample size. This is very common for GWAS summary statistics-based tools. I think it is fine for the R package to require re-naming the columns to fit the format, as one already needs to upload the file into R. However, I find it inconvenient to have to re-save the summary statistics file with different name-columns for the shinyapp tool. Providing e.g. column indexes alone would be much more user-friendly.  Together with the manuscript, I think a longer readme file in the gitlab repository would be very beneficial for stand-alone usage of the R package.Is there enough clear information in the documentation to install, run and test this tool, including information on where to seek help if required?NoAdditional CommentsI cannot answer this question until I can install the tool.Is there a clearly-stated list of dependencies, and is the core functionality of the software documented to a satisfactory level?YesAdditional CommentsHave any claims of performance been sufficiently tested and compared to other commonly-used packages? Not applicableAdditional CommentsThere are no existing comparable tools.Is test data available, either included with the submission or openly available via cited third party sources (e.g. accession numbers, data DOIs)?YesAdditional CommentsAre there (ideally real world) examples demonstrating use of the software? YesAdditional CommentsIs automated testing used or are there manual steps described so that the functionality of the software can be verified?YesAdditional CommentsI can see there is a toy dataset included with the R package.Any Additional Overall Comments to the AuthorI think the manuscript is very clear and good at making the point of the utility of the software. The proposed trumpet plots are very visually appealing and can be useful to characterise the genetic variation of diverse phenotypes. The novelty of the trumpet plots, as compared to previously proposed effect size vs. allele frequency plots, is the use of positive and negative effect sizes, making it look like a trumpet. I also appreciate the style decisions in the standard generated plots, with a nice visually-appealing color scheme and design.  On the use of the software, I have focused my testing on the R package, which I was not able to install. The shinyapp is very useful for visualising the existing, pre-computed trumpet plots, but I do not find it very useful for generating user-uploaded summary statistics for the reasons I mentioned above. Another comment on the ShinyApp is that I appreciate the possibility to download the plots but it would be very useful to include the name of the visualized phenotype as the plot title, for example, to avoid confusion when downloading multiple plots.  I also found an incorrect sentence in the abstract, which is think should be reversed: " The proposed plots have a distinctive trumpet shape, with the majority of variants having low frequency and small effects, while a small number of variants have higher frequency and larger effects".
RecommendationMinor Revisions

---

## [Reviewer Report]

Reviewer name and names of any other individual's who aided in reviewerWentian LiDo you understand and agree to our policy of having open and named reviews, and having your review included with the published manuscript. (If no, please inform the editor that you cannot review this manuscript.)YesIs the language of sufficient quality?YesPlease add additional comments on language quality to clarify if neededIs there a clear statement of need explaining what problems the software is designed to solve and who the target audience is? YesAdditional CommentsIs the source code available, and has an appropriate Open Source Initiative license <a href="https://opensource.org/licenses" target="_blank">(https://opensource.org/licenses)</a> been assigned to the code?YesAdditional CommentsAs Open Source Software are there guidelines on how to contribute, report issues or seek support on the code?YesAdditional CommentsIs the code executable?YesAdditional CommentsIs installation/deployment sufficiently outlined in the paper and documentation, and does it proceed as outlined?YesAdditional CommentsIs the documentation provided clear and user friendly?NoAdditional CommentsMany aspects of Fig.1 are not explained.Additional CommentsIs there a clearly-stated list of dependencies, and is the core functionality of the software documented to a satisfactory level?YesAdditional CommentsHave any claims of performance been sufficiently tested and compared to other commonly-used packages? Not applicableAdditional CommentsAdditional CommentsAre there (ideally real world) examples demonstrating use of the software? YesAdditional CommentsAdditional CommentsAny Additional Overall Comments to the AuthorPlots with allele frequency as x axis and effect size (e.g. odds ratio) as y axis is a very common display of the contribution from both common and rare alleles to genetic association. A schematic  form of this plot is practically on almost everybody's presentation slides when  introduce this topic (to see an example, see, e.g. Science (23 Nov 2012),  vol 338(6110), pp.1016-1017 ). Considering how many people have already been familiar with this type of plot, I feel that very little new is added in this paper: maybe only a new name ("trumpet"), and/or  the power lines. The other methods contributions (log-x, one variant per LD, avoiding gene-level statistics) are rather straightforward. People  without experience with "shiny" (R package) can still use ggplot2  or plot in R to get the same result. Generally speaking, I think the paper is weak, though OK as a program/package announcement.  Major comments:
* I think the trumpet shape (increase of "effect size" for rare variant) is probably a direct consequence of using odds-ratio as a measure of effect size. If the allele frequency in normal population is p0, that in disease population is p1, [p1/(1-p1)]/[p0/(1-p0)] ~ p1/p0 tends to be large for small p0's, simply because the denominator is small. On the other hand, if population attributable risk (p0*(RR-1)/(1+p0*(RR-1)))  is used as the y-axis, I am uncertain what the shape of the plot would be. 
* A risk allele has these pieces of information: 1. allele frequency,  2. effect size (e.g. odds ratio), 3. type-I error/p-value, 4. type-II error/power.  The plot in this paper show #1 vs #2 and #4 being added as extra. In another  publication with a proposal to plot genetic association results (Comp Biol. and Chem. (2014), 48:77-83 doi: 10.1016/j.compbiolchem.2013.02.003),
#2 is against #3 with #1 being an added extra. I'm sure using other combinations  could lead to other types of plots. The authors should discussion/compare  these possibilities.  Minor comments: In Fig.1, the size of the dots, the brown vs cyan color, the discontinuity of  scatter dots around 0.01, are not explained.RecommendationMajor Revisions